# SMART TERNARY QUANTIZATION

## ABSTRACT

Neural network models are resource hungry. Low bit quantization such as binary and ternary quantization is a common approach to alleviate this resource requirements. Ternary quantization provides a more flexible model and often beats binary quantization in terms of accuracy, but doubles memory and increases computation cost. Mixed quantization depth models, on another hand, allows a trade-off between accuracy and memory footprint. In such models, quantization depth is often chosen manually (which is a tiring task), or is tuned using a separate optimization routine (which requires training a quantized network multiple times). Here, we propose Smart Ternary Quantization (STQ) in which we modify the quantization depth directly through an adaptive regularization function, so that we train a model only once. This method jumps between binary and ternary quantization while training. We show its application on image classification.

## 1 INTRODUCTION

Deep Neural Networks (DNN) models have achieved tremendous attraction because of their success on a wide variety of tasks including computer vision, automatic speech recognition, natural language processing, and reinforcement learning (Goodfellow et al., 2016). More specifically, in computer vision DNN have led to a series of breakthrough for image classification (Krizhevsky et al., 2017), (Simonyan & Zisserman, 2014), (Szegedy et al., 2015), and object detection (Redmon et al., 2015), (Liu et al., 2015), (Ren et al., 2015). DNN models are computationally intensive and require large memory to store the model parameters. Computation and storage resource requirement becomes an impediment to deploy such models in many edge devices due to lack of memory, computation power, battery, etc. This motivated the researchers to develop compression techniques to reduce the cost for such models.

Recently, several techniques have been introduced in the literature to solve the storage and computational limitations of edge devices. Among them, quantization methods focus on representing the weights of a neural network in lower precision than the usual 32-bits float representation, saving on the memory footprint of the model. Binary quantization (Courbariaux et al., 2015), (Hubara et al., 2016), (Rastegari et al., 2016), (Zhou et al., 2016), (Lin et al., 2017) represent weights with 1 bit precision and ternary quantization (Lin et al., 2015), (Li & Liu, 2016), (Zhu et al., 2016) with 2 bits precision. While the latter frameworks lead to significant memory reduction compared to their full precision counterpart, they are constrained to quantize the model with 1 bit or 2 bits, on demand. We relax this constraint, and present Smart Ternary Quantization (STQ) that allows mixing 1 bit and 2 bits layers while training the network. Consequently, this approach automatically quantizes weights into binary or ternary depending upon a trainable control parameter. We show that this approach leads to mixed bit precision models that beats ternary networks both in terms of accuracy and memory consumption. Here we only focus on quantizing layers because it is easier to implement layer-wise quantization at inference time after training. However, this method can be adapted for mixed precision training of sub-network, block, filter, or weight easily. To the best of our knowledge this is the first attempt to design a single training algorithm for low-bit mixed precision training.

## 2 RELATED WORK

There are two main components in DNN's, namely, weight and activation. These two components are usually computed in full precision, i.e. floating point 32-bits. This work focuses on quantizing

the weights of the network, i.e. generalizing BinaryConnect (BC) of Courbariaux et al. (2015) and Ternary Weight Network (TWN) of Li & Liu (2016) towards automatic 1 or 2 bits mixed-precision using a single training algorithm.

In BC the real value weights $w$ are binarized to $w^b \in \{-1, +1\}$ during the forward pass. To map a full precision weight to a binary weight, the deterministic sign function is used,

$$w^b = \text{sign}(w) = \begin{cases} +1 & w \geq 0, \\ -1 & w < 0. \end{cases} \tag{1}$$

The derivative of the sign function is zero on $\mathbb{R} \setminus \{0\}$. During back propagation, this cancels out the gradient of the loss with respect to the weights after the sign function. Therefore, those weights cannot get updated. To bypass this problem, Courbariaux et al. (2015) use a clipped straight-through estimator

$$\frac{\partial \mathcal{L}}{\partial w} = \frac{\partial \mathcal{L}}{\partial w^b} \mathbf{1}_{|w| \leq 1}(w) \tag{2}$$

where $\mathcal{L}$ is the loss function and $\mathbf{1}_A(.)$ is the indicator function on the set $A$. In other words (2) approximates the sign function by the linear function $f(x) = x$ within $[-1, +1]$ and by a constant elsewhere. During back propagation, the weights are updated only within $[-1, +1]$. The binarized weights are updated with their corresponding full precision gradients. Rastegari et al. (2016) add a scaling factor to reduce the gap between binary and full-precision model's accuracy, defining Binary Weight Network (BWN). The real value weights $\mathbf{W}$ in each layer are quantized as $\mu \times \{-1, +1\}$ where $\mu = \mathbb{E}\big[|\mathbf{W}|\big] \in \mathbb{R}$. Zhou et al. (2016), generalize the latter work and approximates the full precision weights with more than one bit while Lin et al. (2017) approximate weights with a linear combination of multiple binary weight bases.

## 2.1 TERNARY WEIGHT NETWORKS

Ternary Weight Network (TWN) (Li & Liu, 2016) is a neural network with weights constrained to $\{-1, 0, +1\}$. The weight resolution is reduced from 32 bits to 2 bits, replacing full precision weights with ternary weights. TWN aims to fill the gap between full precision and binary precision weight. Compared to binary weight networks, ternary weight networks have stronger expressive capability. As pointed out in (Li & Liu, 2016), for a $3 \times 3$ weight filter in a convolutional neural network, there is $2^{3 \times 3} = 512$ possible variation with binary precision and $3^{3 \times 3} = 19683$ with ternary precision.

Li & Liu (2016) find the closest ternary weights matrix $\mathbf{W}^t$ to its corresponding real value weight matrix $\mathbf{W}$ using

$$\begin{cases} \hat{\mu}, \hat{\mathbf{W}}^t = \underset{\alpha, \mathbf{W}^t}{\arg\min} \|\mathbf{W} - \mu \mathbf{W}^t\|_2^2, \\ s.t. \ \mu \geq 0, \ w_{ij}^t \in \{-1, 0, 1\}, \ i, j = 1, 2, ..., n. \end{cases} \tag{3}$$

The ternary weight $\mathbf{W}^t$ is achieved by applying a symmetric threshold $\Delta$

$$\mathbf{W}^t = \begin{cases} +1 & w_{ij} > \Delta, \\ 0 & |w_{ij}| \leq \Delta, \\ -1 & w_{ij} < -\Delta. \end{cases} \tag{4}$$

Li & Liu (2016) define a weight dependant threshold $\Delta$ and a scaling factor $\mu$ that approximately solves (3). TWN is trained using stochastic gradient descent. Similar to BC and BWN schemes; ternary-value weights are only used for the forward pass and back propagation, but not for the parameter updates. At inference, the scaling factor can be folded with the input $\mathbf{X}$

$$\mathbf{X} \odot \mathbf{W} \approx \mathbf{X} \odot (\mu \mathbf{W}^t) = (\mu \mathbf{X}) \odot \mathbf{W}^t, \tag{5}$$

where $\odot$ denotes the convolution.

Zhu et al. (2016) proposed a more general ternary method which reduces the precision of weights in neural network to ternary values. However, they quantize the weights to asymmetric values $\{-\mu_1, 0, +\mu_2\}$ using two full-precision scaling coefficients $\mu_1$ and $\mu_2$ for each layer of neural network. While the method achieve better accuracy as opposed to TWN, its hardware implementation becomes a challenge, because there are two unequal full precision scaling factors to deal with.

Our method provides a compromise between BC and TWN and trains weights with a single trainable scaling factor $\mu$. Weights jumps between ternary $\{-\mu, 0, +\mu\}$ and binary $\{-\mu, +\mu\}$. This provides a single algorithm for 1 or 2 bits mixed precision.

## 2.2 REGULARIZATION

Regularization term is the key to prevent over-fitting problem and to obtain robust generalization for unseen data. Standard regularization functions, such as $L_2$ or $L_1$ encourage weights to be concentrated about the origin. However, in case of binary network with binary valued weights, it is more appropriate to have a regularization function to encourage the weights about $\mu \times \{-1, +1\}$, with a scaling factor $\mu > 0$ such as

$$R_1(w, \mu) = \left||w| - \mu\right|, \tag{6}$$

proposed in Belbahri et al. (2019).

A straightforward generalization for ternary quantization is

$$R_2(w, \mu) = \left|\left||w| - \frac{\mu}{2}\right| - \frac{\mu}{2}\right|. \tag{7}$$

Regularizer (6) encourages weights about $\{-\mu, +\mu\}$, and (7) about $\{-\mu, 0, +\mu\}$. The two functions are depicted in Figure 1. These regularization functions are only useful when the quantization depth is set before training start. We propose a more flexible version to smoothly move between these two functions using a shape parameter.

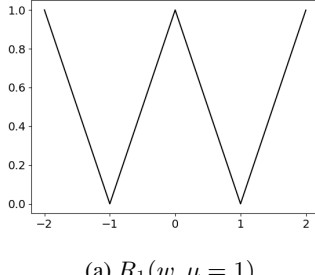
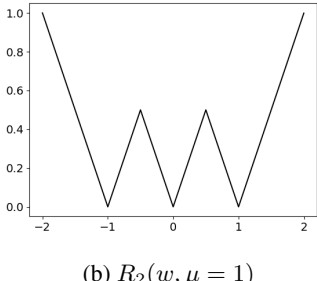

(a) $R_1(w, \mu = 1)$          (b) $R_2(w, \mu = 1)$

Figure 1: Binary and ternary regularizers; $R_1$ encourages binary weights, with minimums at $\{-\mu, +\mu\}$, and $R_2$ encourages ternary weights, with minimums at $\{-\mu, 0, +\mu\}$.

## 3   SMART TERNARY QUANTIZATION

Here we propose an adaptive regularization function that switches between binary regularization of (6) and ternary regularization of (7)

$$R(w, \mu, \beta) = \min\left(\big||w| - \mu\big|, \ \tan(\beta)|w|\right), \tag{8}$$

in which $\mu$ is a trainable scaling factor, and $\beta \in (\frac{\pi}{4}, \frac{\pi}{2})$ controls the transition between (6) and (7). As a special case $\beta \to \frac{\pi}{2}$ converges to the binary regularizer (6) and $\beta \to \frac{\pi}{4}$ coincide with the ternary regularizer (7), depicted in Figure 2. A large value of $\tan(\beta)$ repels estimated weights away from zero thus yielding binary quantization, and small $\tan(\beta)$ values encourage zero weights. The shape parameter $\beta$ controls the quantization depth. Quantization is done per layer, therefore we let $\beta$ very per layer. We recommend to regularize $\beta$ about $\frac{\pi}{2}$ i.e. preferring binary quantization apriori

$$R(w, \mu, \beta) = \min\left(\big||w| - \mu\big|, \ \tan(\beta)|w|\right) + \gamma|\cot(\beta)|, \tag{9}$$

in which $\gamma$ controls the proportion of binary to ternary layers.

For a single filter $\mathbf{W}$ the regularization function is a sum over its elements

$$R(\mathbf{W}, \mu, \beta) = \sum_{i=1}^{I}\sum_{j=1}^{J} \min\left(\big||w_{ij}| - \mu\big|, \ \tan(\beta)|w_{ij}|\right) + \gamma|\cot(\beta)|. \tag{10}$$

Large values of $\gamma$ encourage binary layers. In each layer, weights are pushed to binary or ternary values, depending on the trained value of the corresponding $\beta$. A generalization of (9) towards $L_p$ norms of Belbahri et al. (2019) is also possible. However, here we only focus on regularizer constructed using the $L_1$ norm as the accuracy did not change significantly by using $L_p$ norm with different values of $p$.

The introduced regularization function is added to the empirical loss function $L(.)$. The objective function defined on weights $\mathcal{W}$, scaling factors $\boldsymbol{\mu}$, and quantization depths $\boldsymbol{\beta}$ is optimized using back propagation

$$\mathcal{L}(\mathcal{W}, \boldsymbol{\mu}, \boldsymbol{\beta}) = L(\mathcal{W}) + \sum_{l=1}^{L}\lambda_l \sum_{k=1}^{K_l} R(\mathbf{W}_{kl}, \mu_{kl}, \beta_l), \tag{11}$$

where $k$ indexes the channel, and $l$ indexes the layer. One may use a different regularization constant $\lambda_l$ for each layer to keep the impact of the regularization term balanced across layers, indeed different layers may involve different number of parameters. We set $\lambda_l = \frac{\lambda}{\#\mathbf{W}_l}$ where $\lambda$ is a constant, and $\#\mathbf{W}_l$ is the number of weights in layer $l$.

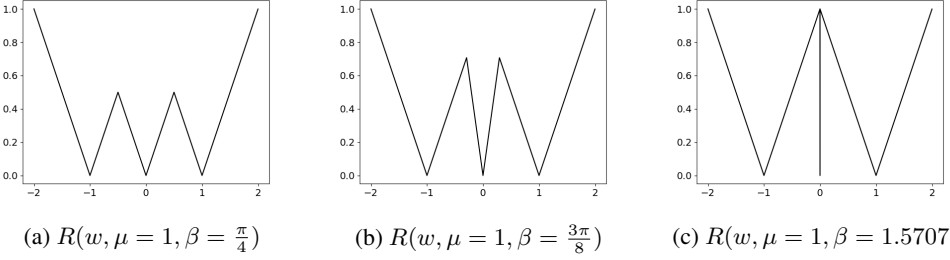

(a) $R(w, \mu = 1, \beta = \frac{\pi}{4})$        (b) $R(w, \mu = 1, \beta = \frac{3\pi}{8})$        (c) $R(w, \mu = 1, \beta = 1.5707)$

Figure 2: Adaptive regularization function. When $\beta \to \frac{\pi}{2}$ the regularization function switches from ternary to binary.

We propose to use the same threshold-based function of Li & Liu (2016) (3), but with a fixed threshold $\Delta_l$ per layer $l$. Note that Li & Liu (2016) propose a weight-dependant threshold. We let the possibility for the weights to only accumulate about $\{-\mu, +\mu\}$ and not about 0, depending on $\beta$.

One may set $\Delta_l$ to have the same balanced weights in $\{-\mu, 0, +\mu\}$ at initialization for all layers and let the weights evolve during training. Formally, if $\sigma_l$ is the standard deviation of the initial Gaussian weights in layer $l$, we propose $\Delta_l = 0.2 \times \sigma_l$. The probability that a single weight lies in the range $[-\Delta_l, \Delta_l]$ is $\approx 0.16$. All the weights falling in this range will be quantized as zeros after applying the threshold function.

Weights are naturally pushed to binary or ternary values depending on $\beta_l$ during training. Eventually, a threshold $\delta$ close to $\frac{\pi}{2} \approx 1.57$ defines the final quantization depth for each layer.

$$\text{Final quantization depth of layer } l : \begin{cases} \text{Binary} & \beta_l \geq \delta, \\ \text{Ternary} & \beta_l < \delta \end{cases}$$

## 4 EXPERIMENT

We run experiments on two simple image classification tasks MNIST (LeCun et al., 1998) and CIFAR10 (Krizhevsky & Hinton, 2009) datasets. We compare our method (STQ) with BinaryConnect (BC) of Courbariaux et al. (2015), Binary Weight Networks (BWN) of Rastegari et al. (2016), Ternary Weights Network (TWN) of Li & Liu (2016) and also with a Full Precision network (FP). The quality of the compression is measured only in terms of memory, it is difficult to compare mixed precision models, with binary and ternary, in terms of consumed energy. Assume that $n_l$ is the quantization depth for the layer $l$ and $\#\mathbf{W}_l$ is the number of weights in layer $l$, therefore the compression ratio is $\frac{\sum_{l=1}^{L} \#\mathbf{W}_l \times 32}{\sum_{l=1}^{L} \#\mathbf{W}_l \times n_l}$. The compression ratio for a binary network is 32, for a ternary network 16, and our approach falls in between.

STQ network generalizes binary and ternary regularization in a single regularization function. First we show how to control the proportion of binary and ternary layers using $\gamma$ in (8). Figure 3 clarifies the effect of $\gamma$ on the weight distribution. When $\gamma$ is large, $\beta$ is encouraged towards $\frac{\pi}{2}$ which corresponds to binary quantization. Consequently, weights are pushed about $\{-\mu, +\mu\}$ and 0 is removed from the trained values, see Figure 3a. On the contrary, when $\gamma$ is small, $\beta$ tends to $\frac{\pi}{4}$ and the weights started including 0 in their values, see Figure 3b.

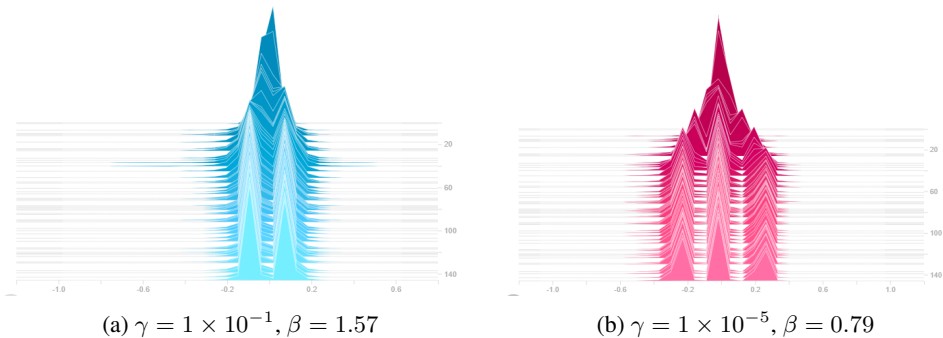

(a) $\gamma = 1 \times 10^{-1}, \beta = 1.57$        (b) $\gamma = 1 \times 10^{-5}, \beta = 0.79$

Figure 3: Effect of $\gamma$ on the weights distribution of a layer while training.

### 4.1 MNIST

MNIST is an image classification benchmark dataset with $28 \times 28$ gray-scales images representing digits ranging from 0 to 9. The dataset is split into 60k training images and 10k testing images. We used the LeNet-5 (LeCun et al., 1998) architecture consisting of 5 layers, 2 convolution followed by maxpooling, stacked with two fully connected layers and a softmax layer at the end. We train the network for 60 epochs using Adam optimizer. We used the initial learning rate of 0.01, but divided by 10 in epoch 15 and 30 to stabilize training. The batch size is set to 64 with $L_2$ weight decay constant $10^{-4}$ only for BC, BWC and TWN. The full precision LeNet-5 is trained with no regularization as it provided a superior accuracy. STQ is trained with $\lambda = 0.1$ and $\gamma = 1 \times 10^{-2}$ and the effective regularization constant is divided by the number of weights in each layer to compensate for the layer size. The best validation accuracy for each method is reported in Table 1, as well as

the quantization depth, and the overall compression ratio. We observe that STQ network quantized the first two convolutional layers in 1 bit, and the last fully-connected layers in 2 bits. The accuracy improvement and the compression ratio is marginal for simple task and simple architectures. The effect of smart training becomes more visible for more complex tasks with more layers.

Table 1: Smart ternary quantization (STQ) compared with Binary Connect (BC), Binary Weight Network (BWN), Terneary Weight Network (TWN), and Full Precision (FP) on MNIST dataset.

|  | Quantization depth per layer (-bits) | Compression ratio | Accuracy (top-1) |
|---|---|---|---|
| BC | 1-1-1-1-1 | 32 | 99.35 |
| BWN | 1-1-1-1-1 | 32 | 99.32 |
| TWN | 2-2-2-2-2 | 16 | 99.38 |
| STQ | 1-1-2-2-2 | 16.3 | 99.37 |
| FP |  | 1 | 99.44 |

## 4.2 CIFAR10

CIFAR10 is an image classification benchmark that contains $32 \times 32$ RGB images from ten classes. The dataset is split into 50k training images and 10k testing images. All images are normalized using mean $= (0.4914, 0.4822, 0.4465)$ and std $= (0.247, 0.243, 0.261)$. For the training session, we pad the sides of the images with 4 pixels, then a $32 \times 32$ crop is sampled, and flipped horizontally at random.

We use two VGG-like architectures, i) VGG-7 architecture defined in Li & Liu (2016) in which we apply batch normalization after each layer and use ReLU activations, ii) a standard VGG-16 architecture. Deep architectures are sensitive to early layer quantization. As commonly practiced, we did not quantize the first and the last layers in VGG-16 for all competing methods.

We train the network for 150 epochs, using Adam optimizer with the initial learning rate 0.001 divided by 10 at epochs 40 and 80. The batch size is set to 64 with $L_2$ weight decay constant $10^{-4}$, moreover $\lambda = 0.1$, $\gamma = 10^{-3}$ for STQ. The best validation accuracy for each method is reported in Table 2. STQ beats pure 2 bits network TWN, even in terms of accuracy. It recommends three 1 bit layers for VGG-7 and seven 1 bit layers for VGG-16. The compression ratio is significantly higher than a ternary network. The weight distribution of each layers are depicted in Figure 4 for the VGG-7 architecture. Weights are pushed to $\{-\mu, +\mu\}$ or $\{-\mu, 0, +\mu\}$ depending on the shape parameter $\beta$.

Table 2: Smart ternary quantization (STQ) compared with Binary Connect (BC), Binary Weight Network (BWN), Terneary Weight Network (TWN), and Full Precision (FP) on CIFAR10 dataset.

| Architecture | Method | Quantization depth per layer (-bits) | Compression ratio | Accuracy (top-1) |
|---|---|---|---|---|
| VGG-7 | BC | 1-1-1-1-1-1-1 | 32 | 92.49 |
|  | BWN | 1-1-1-1-1-1-1 | 32 | 92.42 |
|  | TWN | 2-2-2-2-2-2-2 | 16 | 92.74 |
|  | STQ | 2-1-1-1-2-2-2 | 18.3 | **92.94** |
|  | FP |  | 1 | 93.72 |
| VGG-16 | BC | 32-1-1-1-1-1-1-1-1-1-1-1-1-32 | 31.5 | 91.92 |
|  | BWN | 32-1-1-1-1-1-1-1-1-1-1-1-1-32 | 31.5 | 91.85 |
|  | TWN | 32-2-2-2-2-2-2-2-2-2-2-2-2-32 | 15.9 | 92.14 |
|  | STQ | 32-2-1-1-2-2-2-1-1-1-1-1-2-32 | 25.1 | **92.38** |
|  | FP |  | 1 | 92.53 |

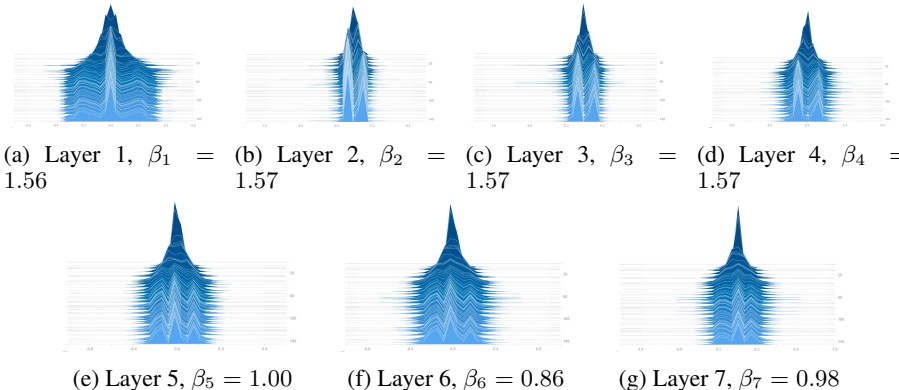

(a) Layer 1, $\beta_1$ = 1.56    (b) Layer 2, $\beta_2$ = 1.57    (c) Layer 3, $\beta_3$ = 1.57    (d) Layer 4, $\beta_4$ = 1.57

(e) Layer 5, $\beta_5 = 1.00$    (f) Layer 6, $\beta_6 = 0.86$    (g) Layer 7, $\beta_7 = 0.98$

Figure 4: Layer wise weights distribution in VGG-7 for STQ. The weights are pushed to binary when the shape parameter $\beta$ is close to $\frac{\pi}{2} \approx 1.57$.

## 5 CONCLUSION

Smart ternary quantization (STQ) is a training method to build a 1 and 2 bits mixed quantized layers. Depth optimization requires training network multiple times which is costly, specially if the network is complex. This approach successfully combines quantization with different depths, while training the network only once. We tried layer-wise quantization, because it is more suitable for mixed-precision inference implementation. However, subnetwork, block, filter, or weight mixed quantization is feasible using a similar algorithm.

STQ makes manual tuning of quantization depth unnecessary. It allows to improve the memory consumption, by automatically quantizing some layers with smaller precision. This method sometimes even outperforms pure ternary networks in terms of accuracy thank to a formal regularization function that shapes trained weights towards mixed-precision.

It is well-known that some layers are more resilient to aggressive quantization. Our introduced methodology trains network similar to pure ternary but give an insight about which layers can be simplified further by going to binary quantization depth.

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
