# OpenReview forum: "Smart Ternary Quantization"
_ICLR.cc/2020/Conference — Reject_

### Official Review · AnonReviewer3 · 2019-10-16
**Official Blind Review #3**

**Rating:** 6

**Review:**

The paper discusses a generalization to low bit quantization and combines the approaches of binary and ternary quantization methods. Past methods such as Binary Connect and Binary Weights Network have shown that you can train a network efficiently with 1-bit quantization, and methods such as Ternary Weights Network demonstrate 2-bit quantization with weights taking one of {-1, 0, 1} * mu, with mu being a scale computed per weight tensor. The authors generalize these two methods so that the choice of binary vs ternary weights can be made per layer automatically during training. The primary contribution to make that work is by incorporating a generic regularizer with addition hyper-parameters to trade-off between the binary weight regularizer and ternary weight regularizer. In addition to that, the regularization also includes a prior to make the layers prefer binary weights by default. This is done by adding a cost that penalizes the choice of ternary weights for each layer.

Overall, the paper is well written and explained, with supporting experiments to show on MNIST and on CIFAR10 that this method performs quite competitively compared to an all-binary or all-ternary weights network. Low-bit quantization is an important research area and this paper makes a strong contribution by studying mixed-precision low-bit quantization. The mathematical explanations of the generalized regularizer are well justified. For instance, the regularization constant lambda is normalized for each layer by the total number of weights to evenly weight all layers.

Although the experiments cover MNIST and CIFAR10, it's not clear how mixed-precision low-bit methods perform on models more prevalent in the real-world. Supporting the experiments with ResNet variants on ImageNet would help clarify that further.  The paper very well explains the fundamentals of 1-bit and 2-bit methods, and the contribution of this paper (generalization of these two methods) is a somewhat natural extension without significant novelty. Moreover, in addition to the accuracy, it would also be better to understand how the run-time performance of such models (on existing software and hardware implementations) compares to pure binary and ternary networks, as knowledge of that would reveal insights into systems optimizations to be made in future work in this field. Given all of this, my rating is a weak accept.

Pros:
- The problem and the fundamentals (prior work) are very well laid out.
- The regularization component that generalizes the two types of quantization is sound.
- Experiments on CIFAR10 strongly show improved accuracy and higher compression ratio compared to ternary weights network.

Cons:
- Lacking more realistic experiments on larger datasets such as ImageNet and models like ResNets.
- The runtime performance of how STQ compares to 1-bit and 2-bit quantization variants isn't shown.
- The regularizer introduces more hyper-parameters to tweak and it's not clear how sensitive these are to the choice of the architecture. Different values of gamma are used in the two experiments, and further analysis on how the performance varies for various values of lambda and gamma would shed further insight.

Minor comments:
- In equation (3), the term under argmin should be mu and not alpha.
- Section 3, line above equation (9) reads "very per layer" instead of "vary per layer"

**Experience Assessment:**

I have published one or two papers in this area.

**Review Assessment: Checking Correctness Of Derivations And Theory:**

I carefully checked the derivations and theory.

**Review Assessment: Checking Correctness Of Experiments:**

I assessed the sensibility of the experiments.

**Review Assessment: Thoroughness In Paper Reading:**

I read the paper at least twice and used my best judgement in assessing the paper.

---

> ### Author Response · Authors · 2019-11-15
> **Answer to Official Blind Review #3**
>
> Thank you for your comments. We are running experiments for larger datasets and more architectures.
>
> We will provide run-time performance disucssion in the conclusion section.
>
> We will add sensitivity study about these parameters in the next version or on our github codes.
>
> Minor comments will be corrected in the next version of the paper.

---

### Official Review · AnonReviewer2 · 2019-10-22
**Official Blind Review #2**

**Rating:** 1

**Review:**

This paper presents an approach where the regularisation is used to optimise whether each layer of a DNN is binary or ternary. The paper presents an equation for performing this along with two examples of the process in use.

The paper is inconsistently written with work described at different levels in different sections and has an inconsistent feeling about it. For example the introduction seems to stop abruptly before it describes all the parts of the paper.

The paper seems to contain an idea which might have merit. However, the idea just does not seem to have been developed enough.

Major concerns:
1) The authors claim that this is the first attempt at a training algorithm for mixed precision training. However, a simple google search throws up many papers in this area. Many of which are not mentioned in the related work.

2) Equations are not discussed in enough detail, nor are the parameters defined. Or if they are defined they are done so much later in the work.

3) There doesn’t seem to be enough material here to reproduce the work.

4) In the results you talk about ‘the best’. Given that there has been much criticism over the last two years about good academic practice the fact that you don’t say at least ‘best out of …’ is worrying.

5) You have magic parameters lambda and gamma. You say that these effect the outcome of the work but in your examples you only state values these are set to. One would expect to see at least some analysis of how varying these values effect the outcome. But better would be to show that you have identified good values for both of these parameters. Ideally would be an evaluation of how others could identify the best values.

6) Figures 3 and 4 are difficult to interpret. They need a clear explanation.

Some more generic comments:
- The abstract seems to assume a huge prior knowledge by the reader.

- ‘1 bit precision’ - precision seems to have no meaning in this context. Surely just ‘1 bit’

- The related work contains a lot of equations, but no real explanation of what they are.

- ‘we let β very per layer’ -> ‘we let β vary per layer’

- In equation 10 what do I and J represent?

- ’28 × 28 gray-scales images’ -> ’28 × 28 gray-scale images’

- ‘For the training session, we pad the sides of the images with 4 pixels, then a 32 × 32 crop is sampled, and flipped horizontally at random.’ - why?

- ‘As commonly practiced’ - by whom?

- ‘which is costly, specially if’ -> ‘which is costly, especially if’


**Experience Assessment:**

I have read many papers in this area.

**Review Assessment: Checking Correctness Of Derivations And Theory:**

I assessed the sensibility of the derivations and theory.

**Review Assessment: Checking Correctness Of Experiments:**

I assessed the sensibility of the experiments.

**Review Assessment: Thoroughness In Paper Reading:**

I read the paper thoroughly.

---

> ### Author Response · Authors · 2019-11-15
> **Answer to Official Blind Review #2**
>
> 1)  We believe that the reviewer missed the main point of the paper here. Most of the works that have been published on mixed precision training assumed that the quantization depth is known before training. Other works that optimize quantization depth use a separate optimizer. The novelty of our approach lies in the fact that the neural network learns quantization depth for each layer only by a modifed  back-propagation. We modify back-propagation only by adding a regularization function, and this allows to train weights, and to estimate quantization depth simultaneously.
>
> 2)  We will be looking at parameter definition and interpretation again. We only introduced two new parameters gamma, and beta in which we provided interpretation.  Other equations are only the definition of regularization function and they are discussed in detail in reference [16].
>
> 3) This is a high-level comment.  We did our best to provide enough details to reproduce the work. We can improve the descriptions if the missing points are pinpointed. We provide a github code source after acceptance of the paper.
>
> 4) We agree. We must avoid using the word 'the best' in scientific articles as 'the best' may change through time. We will correct this in the next version.
>
> 5) We appreciate this detailed comment. We will add some studies about varying lambda and gamma parameters in the next version, or on our github codes.
>
> 6) The point of these figures is to see that the weights are indeed pushed on binary or ternary values depending on the shape parameter beta of the regularization function. More clear explanations will be added in the next version of the paper, see reference [16].
>
> Thanks for the generic comments. They will be considered for the revised version.

---

### Official Review · AnonReviewer1 · 2019-10-26
**Official Blind Review #1**

**Rating:** 3

**Review:**

The Paper talks about the Smart Ternary Quantization method that improves the quantization over binary and ternary quantizations by specifying an adaptive quantization. The proposed regularization function is covered in detail and the results are evaluated on MNIST and CIFAR10 datasets

The authors can improve the submission by
1. evaluating more modern networks with bigger datasets, as opposed to the ones demonstrated.
2. describing gamma (eq 10) , it wasn't clear why that parameter was introduced (in addition to the beta) and it's significance, what was more confusing is coverage for this instead of beta in the experimental setup
3. why the LR needed to be modified for the described method

**Experience Assessment:**

I have read many papers in this area.

**Review Assessment: Checking Correctness Of Derivations And Theory:**

I did not assess the derivations or theory.

**Review Assessment: Checking Correctness Of Experiments:**

I carefully checked the experiments.

**Review Assessment: Thoroughness In Paper Reading:**

I read the paper thoroughly.

---

> ### Author Response · Authors · 2019-11-15
> **Answer to Official Blind Review #1**
>
> 1. We agree that experiments on networks and datasets are minimal. The main goal of this paper was to demonstrate that it is possible to train a neural network that jump  between 1-bit or 2-bits while only adding a regularization function to the loss. In the litterature, mostly tuning quantization depth is performed by an independent optimization algorithm like Bayesian Optimization. This is the first attempt to modify back-propagation using only a regularizer that it tunes quantization depth mutually while training.
>
> 2. Gamma is a hyperparameter which is fixed before training and controls how aggressive the quantization is towards binary weights. Beta is a trainable parameter which modify the shape of the regularization function and enforce weights to be either binary or ternary. The trained beta values, quantify the relative proportion of binary weights to ternary weights. We did not cover beta in the experimental set up because its value change during training.
>
> 3.  The LR was not modified for the described method. LR was changed between the two experiments (LR = 0.01 for MNIST, LR = 0.001 for CIFAR10) to achieve competitive accuracy.

---

### Official Review · AnonReviewer4 · 2019-12-05
**Official Blind Review #4**

**Rating:** 3

**Review:**

This paper studies mixed-precision quantization in deep networks where each layer can be either binarized or ternarized. The authors propose  an adaptive regularization function that can be pushed to either 2-bit or 3-bit through different parameterization, in order to automatically determine the precision of each layer. Experiments are performed on small-scale image classification data sets MNIST and CIFAR-10.

The proposed regularization method is simple and straightforward. However, many details are not stated clearly enough for reproduction. E.g, since the proposed regularization already promotes binary or ternary weights, whey is there still a thresholding operation at the end of Section 3? Is it because the proposed regularization can not provide strict binary or ternary weights? Does the method require one more hard binarization/ternarization step after \beta is learned. Indeed, tan(x) is not well-defined when x=pi/2, and the derivative tan'(x)= 1+tan^2(x) can be large when x is near pi/2, and does gradient descent work well in this case?

The experiments are only performed on small-scale data sets. Thus it is hard to tell if the proposed method also works for larger networks or data sets? Moreover, it is not fair to use "best validation accuracy" for comparison with other methods since the validation set is seen during training and it is not clear if the hyper-parameters of the proposed methods are tuned for best performance on the seen validation set. It would be more fair to compare the test accuracy like in the BinaryConnect (BC) paper. Yet another concern is that many recent methods that can train mixed-precision networks are not compared. For instance, the HAQ method [1] searches for precision for each layer using the reinforcement learning method, how does the proposed method perform when compared with it?

[1]. Wang, Kuan, et al. "HAQ: Hardware-Aware Automated Quantization with Mixed Precision." Proceedings of the IEEE Conference on Computer Vision and Pattern Recognition. 2019.


**Experience Assessment:**

I have published one or two papers in this area.

**Review Assessment: Checking Correctness Of Derivations And Theory:**

I assessed the sensibility of the derivations and theory.

**Review Assessment: Checking Correctness Of Experiments:**

I assessed the sensibility of the experiments.

**Review Assessment: Thoroughness In Paper Reading:**

I read the paper at least twice and used my best judgement in assessing the paper.

---

### Public Comment · ~Lukas_Enderich1 · 2019-11-07
**Request regarding the experimental evaluation, especially concerning the accuracy values in the comparisons.**

The authors present their work on learning mixed precision bit-sizes for quantization of neural networks. The article is well structured and well written, so it is very easy to read and easy to follow. We are working in the same domain of network quantization, have also published articles and therefore we would like the authors to comment on the following points:

1) We are slightly confused about the accuracy values presented in section 4.1 and 4.2. For example, you compare with Ternary Weight Networks (TWN, Li & Liu 2016) and mention 92.74% accuracy on CIFAR-10 (VGG-7) and 99.38% on MNIST, respectively. However, the accuracy values presented in the original TWN paper are 92.56% and 99.35%, respectively. The same applies for BinaryConnect (BC, 2015). So, we are wondering how the authors came up with the mentioned results (which differ from the values reported in the original papers). Did you use your own implementation? Please, explain in more detail.

2) The most recent comparison in your paper, which is the comparison with TWN, originates from 2016.  From 2016 to 2019 several new approaches for quantization have been published. Therefore, we would like the authors to compare their method with the most recent methods (for example, Variational Network Quantization [1] or Explicit Loss-error-aware Quantization [2]).  Moreover, we published a paper about ternary-valued networks at the ESANN 2019 [3] - we used the same VGG-7 architecture on CIFAR-10 and performed clearly better than TWN (93.73%). We would really appreciate if the authors can also compare with our results.

[1] Jan Achterhold, Jan Mathias Koehler, Anke Schmeink, and Tim Genewein. Variational network quantization. ICLR 2018.

[2] Aojun Zhou, Anbang Yao, Kuan Wang, and Yurong Chen. Explicit loss-error-aware quantization for low-bit deep neural networks. CVPR 2018.

[3] Lukas Enderich, Fabian Timm, Lars Rosenbaum, and Wolfram Burgard. Learning Multimodal Fixed-Point Weights usingGradient Descent. ESANN 2019.

3) Additionally, it would be really nice if the authors could compare their method with smaller and more optimized networks like ResNet-56 oder DenseNet (both less than 1M parameters) to investigate whether their approach also yields good results even for networks that a difficult in terms of quantization.

---

### Decision · Program_Chairs · 2019-12-19

**Decision:**

Reject

**Comment:**

This paper studies mixed-precision quantization in deep networks where each layer can be either binarized or ternarized. The proposed regularization method is simple and straightforward. However, many details and equations are not stated clearly. Experiments are performed on small-scale image classification data sets. It will also be more convincing to try larger networks or data sets. More importantly, many recent methods that can train mixed-precision networks are not cited nor compared. Figures 3 and 4 are difficult to interpret, and sensitivity on the new hyper-parameters should be studied. The use of "best validation accuracy" as performance metric may not be fair. Finally, writing can be improved. Overall, the proposed idea might have merit, but does not seem to have been developed enough.